# Brief communication: Stalagmite damage by cave-ice flow quantitatively assessed by fluid-structure-interaction simulations

Alexander H. Jarosch[1], Paul Hofer[2], and Christoph Spötl[3]

[1]ThetaFrame Solutions, Kufstein, 6330, Austria
[2]Institute of Basic Sciences in Engineering Sciences, University of Innsbruck, 6020, Innsbruck, Austria
[3]Institute of Geology, University of Innsbruck, 6020, Innsbruck, Austria

**Correspondence:** Alexander H. Jarosch (research@alexj.at)

**Abstract.** Mechanical damage to stalagmites is commonly observed in mid-latitude caves. Former studies identified thermoelastic ice expansion as a plausible mechanism for such damage. This study builds on these findings and investigates the role of ice flow along the cave bed as a possible second mechanism for stalagmite damage. Utilizing fluid-structure interaction models based on the finite element method, forces created by ice flow are simulated for different stalagmite geometries. The resulting effects of such forces on the structural integrity of stalagmites are analyzed and presented. Our results suggest that structural failure of stalagmites caused by ice flow is possible, albeit unlikely.

## 1 Introduction

Secondary carbonate deposits such as stalactites and stalagmites, collectively referred to as speleothems, are abundant in caves worldwide and grow over extended periods of time, typically many thousands of years, thereby recording valuable proxy information about climate and vegetation outside the cave. Speleothems are therefore highly sought archives of paleoenvironmental change (e.g. Fairchild and Baker, 2012). Despite the fact that caves are typically well protected from processes such as weathering or erosion, some caves particularly in the mid-latitudes show naturally occurring, mechanical damage to speleothems. Several processes have been discussed to cause damage to stalagmites and stalactites, including movement of stalagmites growing on a soft-sediment substrate, gravity-driven breakdown of cave ceilings, strong earthquakes, and the former presence of ice in these subterranean cavities. The latter process has been identified as a main cause for stalagmites that were sheared off their base (e.g. Gilli, 2004; Orvošová et al., 2012), as well as for sheet-like deposits known as flowstones showing a fractured appearance (e.g. Lundberg and McFarlane, 2012, see Fig. 1). Key to this interpretation is the coexistence of damaged speleothems with so-called cryogenic cave carbonates (Richter et al., 2011; Zak et al., 2018; Spötl et al., 2021). The former are indicators of the past presence of perennial ice in these caves and can be reliably dated using radiometric techniques (e.g. Dublyansky et al., 2024). In addition, damaged speleothems are often overgrown by younger calcite of Holocene age, which proves the glacial age of the damage (e.g. Zak et al., 2018; Spötl et al., 2021) In a recent study, Spötl et al. (2023) used numerical modelling combined with laboratory measurements to quantitatively assess which processes in former ice-filled caves led to damage of stalagmites. The results show that internal deformation within a cave ice body due to ice flow under gravity cannot fracture stalagmites, even on steep slopes. In contrast, thermoelastic stresses within an ice body due to temperature changes reach values

partly exceeding the tensile strength of stalagmites. In this contribution we expand on the analysis of Spötl et al. (2023) by (a) examining not only the external forces exerted by ice flow but also their internal effect within stalagmites, and (b) including also smaller stalagmite dimensions. Structural failure of stalagmites caused by ice flow was investigated using a fluid structure interaction (FSI) model (e.g. Sigrist, 2015).

## 2  Methods

In order to be consistent with the study of Spötl et al. (2023), we chose to model the same ice domain with $-7 \leq x \leq 14$ m, $-3.5 \leq y \leq 3.5$ m and the stalagmite location at $x = y = 0.0$ m (cf. Fig. 3 in Spötl et al., 2023). However, we examined two ice thicknesses ($h_{\mathrm{ice}} = 1.0$ and $2.0$ m), two slope angle values ($\beta_{\mathrm{ice}} = 10$ and $40°$), three stalagmite diameters ($d_s = 0.15, 0.25$ and $0.4$ m), two stalagmite heights ($h_s = 0.5$ and $1.0$ m) as well as full-slip and no-slip conditions at the cave base. We do not utilize any glaciological sliding laws, as we want to investigate the two end-member states (frictionless full sliding vs.

no-slip). In total, 14 different configurations which are summarized in Tab. 2 were examined. The cave walls (sidewalls of the ice domain) were assumed to be no-slip boundary conditions and the ice surface is a stress-free boundary, allowing for free movement of ice. Stalagmites were modelled as cylinders with a fixed diameter $d_s$, except for configurations 13 and 14, which exhibit a conical shape ($d_s^{\mathrm{base}} = 0.25$ m and $d_s^{\mathrm{top}} = 0.1$ m). Our numerical mesh contains typically 100.000 tetrahedral elements with mesh element sizes varying between 0.3 m far away from the stalagmite to 0.02 m within the stalagmite.

## 40  2.1  FSI Model

We assumed the deformation of ice to be fluid like, with a non-Newtonian rheology following Glen's flow law (e.g. Glen, 1955). Driven by gravity, the cave ice flows downslope and exerts forces on the stalagmite's outer surface. The structural response of the stalagmite to these external forces was modelled as linearly elastic. In Spötl et al. (2023), laboratory experiment results have been presented identifying the elastic properties of the stalagmite material. As we are interested in the stress distribution

just before structural failure, linear elasticity is a reasonable choice for our model (e.g. Zienkiewicz et al., 2013). Using the chosen FSI model approach we simulated both, the forces exerted by ice deformation (fluid) and the response of the stalagmite (solid structure) to these forces with a finite-element model. A summary of the relevant model parameters is given in Tab. 1.

We developed this simulation strategy using the open-source software *Elmer FEM* (e.g. Malinen and Råback, 2013), which includes well established FSI routines (e.g. Järvinen et al., 2008). To include the non-linear flow behaviour of ice in our model,

we used *Elmer/Ice* (e.g. Gagliardini et al., 2013), which builds upon *Elmer FEM*.

*Elmer FEM* applies a partitioned approach to solving FSI problems, thus solving the equations governing non-linear fluid flow and elastic structural response with two different numerical solvers. Forces are coupled at the fluid-structure boundary (stalagmite outer surface) and the model iterates until a balance between fluid stress and elastic response of the solid is found.

## 3 Results

Solutions from the FSI simulations were analyzed for maximum (principal) tensile stresses $\sigma_{\mathrm{max}}$ in the stalagmite to identify cases where its tensile strength ($4.3 \pm 1.5 \times 10^6 \, \mathrm{Pa}$; Spötl et al., 2023) is exceeded. According to Rankine's theory (Rankine, 1857) this would result in structural failure of the stalagmite. Typically, the maximum tensile stresses were found to be caused by bending of the stalagmite and occurred at the upstream side of its base in vertical direction (s. Fig. 2). The results of the simulations are summarized in Tab. 2.

For moderately sloping caves ($\beta_{\mathrm{ice}} = 10°$), only configurations exhibiting full-slip conditions at the ice-cave base interface (no bed friction) indicate failure (simulations 04, 06, 08, and 14). In case of steep sloping caves ($\beta_{\mathrm{ice}} = 40°$), all investigated configurations (simulations 09 to 12) exhibit maximum tensile stresses significantly exceeding the stalagmite's tensile strength.

We also investigated the effect of ice temperature, as all our simulations assume temperate ice (cf. $A$ in Tab. 1). However, reducing the ice temperature to $-5.0°$ C only results in a reduction of the maximum tensile stress of less than $0.5 \, ‰$ relative
to the temperate case.

Ice thickness $h_{\mathrm{ice}}$, and thereby how much ice overflows the stalagmite, significantly affects $\sigma_{\mathrm{max}}$. Comparing simulations 05 ($h_{\mathrm{ice}} = 2 \, \mathrm{m}$) and 07 ($h_{\mathrm{ice}} = 1 \, \mathrm{m}$) reveals a reduction of $\sigma_{\mathrm{max}}$ by about 45 % in case of no-slip conditions. In case of full-slip conditions (simulations 06 and 08), however, only a small reduction of $\sigma_{\mathrm{max}}$ by about 8 % for the same $50\%$ reduction in $h_{\mathrm{ice}}$ is observed.

The stalagmite geometry is found to significantly influence the observed maximum tensile stresses. This is especially true for the stalagmite base diameter $d_s$, and by extension its section modulus $S = \frac{\pi d_s^3}{32}$. Even though a reduction of $d_s$ reduces the surface area exposed to ice flow, $\sigma_{\mathrm{max}}$ for simulations 03 and 04 ($d_s = 0.25 \, \mathrm{m}$) is nearly three times higher than for simulations 01 and 02 ($d_s = 0.4 \, \mathrm{m}$) due to a disproportionate reduction of $S$. Given the same ice thickness of $h_{\mathrm{ice}} = 2 \, \mathrm{m}$, small stalagmites ($d_s = 0.15 \, \mathrm{m}$ and $h_s = 0.5 \, \mathrm{m}$; simulations 05 and 06) are more prone to failure than larger stalagmites ($d_s = 0.4 \, \mathrm{m}$
and $h_s = 1 \, \mathrm{m}$; simulations 01 and 02). Conically shaped stalagmites (configurations 13 and 14) exhibit comparable, albeit slightly lower, maximum tensile stresses than cylindrical stalagmites (configurations 03 and 04).

For our simulations we assumed ice frozen to the stalagmite and thus a direct force coupling on the ice-stalagmite interface. Therefore, the modelled ice flow can potentially exert tensile forces on the downslope side of the stalagmite. By comparing the maximum principle tensile stress $\sigma_{\mathrm{max}}$ on the downslope ice-stalagmite interface with reported values for ice tensile strength
($0.7 - 3.1 \times 10^6 \, \mathrm{Pa}$ (Petrovic, 2003)), we identified two configurations (configurations 10 and 12) for which failure of the ice itself would likely occur. For these configurations the reported maximum $\sigma_{\mathrm{max}}$ values are definitely an overestimation as they assume full stress coupling between the ice and stalagmite on the downslope side. In reality, at such high stresses, the ice itself would fail and cracks would form which would decouple the ice body from the stalagmite at the downstream side. Our simulations do not account for such crack formations and thus the results for configurations 10 and 12 have to be considered
with caution.

High shear stresses within the stalagmite could also lead to structural failure, hence we report maximum absolute values of the base shear stress $|\tau_{zx}|$ in Tab. 2. However, maximum $|\tau_{zx}|$ amounts to less than 30 % of maximum $\sigma_{\mathrm{max}}$ for all investigated configurations. We therefore consider it unlikely that the stalagmite will fail due to the presence of shear stresses.

## 4 Discussion and conclusion

Ice caves exhibit complex geometries which comprise galleries, chambers and shafts. In addition, the floor of chambers and galleries is typically covered by coarse debris including variably large and angular blocks. These topographic intricate features render free sliding (i.e. full-slip) interface conditions between the ice and the cave bed highly unlikely in reality. Most importantly, cave ice bodies are typically frozen to their substrate. They owe their existence to the strong cooling of the rock mass surrounding the cave as a consequence of strong (winter) cooling by air flow or the presence of permafrost (e.g. Luetscher, 95 2022).

Any zones of full-slip would get compensated by ice being required to flow around and/or over complex bed topographies. Hence we consider our results with basal full-slip conditions (simulation numbers 02, 04, 06, 08, 10, 12, and 14) as highly unlikely upper end member cases for ice cave conditions. Conditions close to the no-slip simulations are regarded as most likely.

We found large tensile stresses on the upslope (i.e. upstream) side of stalagmites to be the main cause for structural failure. In contrast, a large influence of shear stresses near the stalagmite base is ruled out due to their comparatively small magnitude of less than 30 % of the observed tensile stresses.

Our results suggest that only in steep sloping caves with large ice bodies (e.g. 14 m of cave ice, 2 m thick, upslope of a stalagmite), forces exerted by ice flow can become large enough to damage moderately large stalagmites. Only in one case with 105 no-slip conditions (simulation number 09) we find a configuration in which a moderately large stalagmite would be damaged.

This study confirms that vertical uplift of stalagmites due to thermoelastic stresses induced by expanding ice, as previously reported (Spötl et al., 2023), is still the most likely physical explanation for wide-spread damage to speleothems in mid-latitude caves during the cold periods of the Pleistocene. Even though our results rule out lateral ice flow as a main damage mechanism, which was previously often assumed, in extreme conditions (i.e. steep sloping, large, ice filled caves) however, large tensile 110 stresses caused by ice flow can occur in stalagmites, which ultimately can also cause structural damage.

*Code availability.* All fluid structure interaction simulations in this contribution were carried out with the *Elmer FEM* and *Elmer/Ice* software packages, which can be accessed online under http://elmerfem.org and http://elmerice.elmerfem.org/

*Author contributions.* CS and AHJ conceived the study. AHJ performed the numerical simulations and together with PH analyzed the results. All authors contributed to writing the paper.

*Competing interests.* The contact author has declared that none of the authors has any competing interests.

*Acknowledgements.* We would like to thank the two anonymous reviewers as well as the editor in charge for their work during the publication process. This research was funded in part by the Austrian Science Fund (FWF), grants P318740 and P358770.

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

**Table 1.** Model parameters

| Symbol | Value | Units | Description |
| --- | --- | --- | --- |
| $A$ | $2.4 \times 10^{-24}$ | $\mathrm{Pa}^{-3}\,\mathrm{s}^{-1}$ | Glen's flow law parameter for temperate ice |
| $n$ | 3 | - | Glen's flow law non-linearity number |
| $E_i$ | $9.5 \times 10^{9}$ | Pa | Young's modulus for ice |
| $E_s$ | $6.41 \times 10^{10}$ | Pa | Young's modulus for stalagmite |
| $g$ | 9.81 | $\mathrm{m\,s}^{-2}$ | gravitational acceleration |
| $\nu_i$ | 0.33 | - | Poisson ratio for ice |
| $\nu_s$ | 0.27 | - | Poisson ratio for stalagmite |
| $\rho_i$ | 910 | $\mathrm{kg\,m}^{-3}$ | Density of ice |
| $\rho_s$ | 2670 | $\mathrm{kg\,m}^{-3}$ | Density of stalagmite material |

Zienkiewicz, O., Taylor, R., and Zhu, J.: The Finite Element Method: Its Basis and Fundamentals, Butterworth-Heinemann, 2013.

**Table 2.** Maximum principal stress max. $\sigma_{\mathrm{max}}$ and maximum shear stress max. $|\tau_{zx}|$ for all simulations. $\beta_{\mathrm{ice}}$ is the cave slope angle, $h_{\mathrm{ice}}$ the ice thickness, $h_s$ the stalagmite height, $d_s$ the stalagmite diameter. Boundary conditions at the ice bed are listed as well. Bold simulation numbers indicate configurations for which the stalagmite would likely break due to the arising tensile stresses. The configurations exhibiting a conical stalagmite shape are marked with an asterisk (*). For the simulations marked with a plus sign (+), maximum $\sigma_{\mathrm{max}}$ on the downslope side of the stalagmite exceeds the tensile strength of ice (cf. Sect. 3).

| Sim. № | $\beta_{\mathrm{ice}}$ | $h_{\mathrm{ice}}$ | $h_s$ | $d_s$ | base BC | max. $\sigma_{\mathrm{max}}$ | max. $|\tau_{zx}|$ |
|---|---|---|---|---|---|---|---|
| | ° | m | m | m | | $10^6\,\mathrm{Pa}$ | $10^6\,\mathrm{Pa}$ |
| 01 | 10 | 2 | 1 | 0.4 | no-slip | 1.22 | 0.273 |
| 02 | 10 | 2 | 1 | 0.4 | slip | 3.53 | 0.789 |
| 03 | 10 | 2 | 1 | 0.25 | no-slip | 3.55 | 0.745 |
| **04** | 10 | 2 | 1 | 0.25 | slip | 11.1 | 2.36 |
| 05 | 10 | 2 | 0.5 | 0.15 | no-slip | 2.73 | 0.588 |
| **06** | 10 | 2 | 0.5 | 0.15 | slip | 15.1 | 3.27 |
| 07 | 10 | 1 | 0.5 | 0.15 | no-slip | 1.51 | 0.325 |
| **08** | 10 | 1 | 0.5 | 0.15 | slip | 13.9 | 3.01 |
| **09** | 40 | 2 | 1 | 0.25 | no-slip | 26.1 | 6.58 |
| **10+** | 40 | 2 | 1 | 0.25 | slip | 86.9 | 22.2 |
| **11** | 40 | 2 | 0.5 | 0.15 | no-slip | 22.4 | 5.75 |
| **12+** | 40 | 2 | 0.5 | 0.15 | slip | 105 | 27.6 |
| 13* | 10 | 2 | 1 | 0.25 | no-slip | 3.09 | 0.712 |
| **14*** | 10 | 2 | 1 | 0.25 | slip | 10.3 | 2.4 |

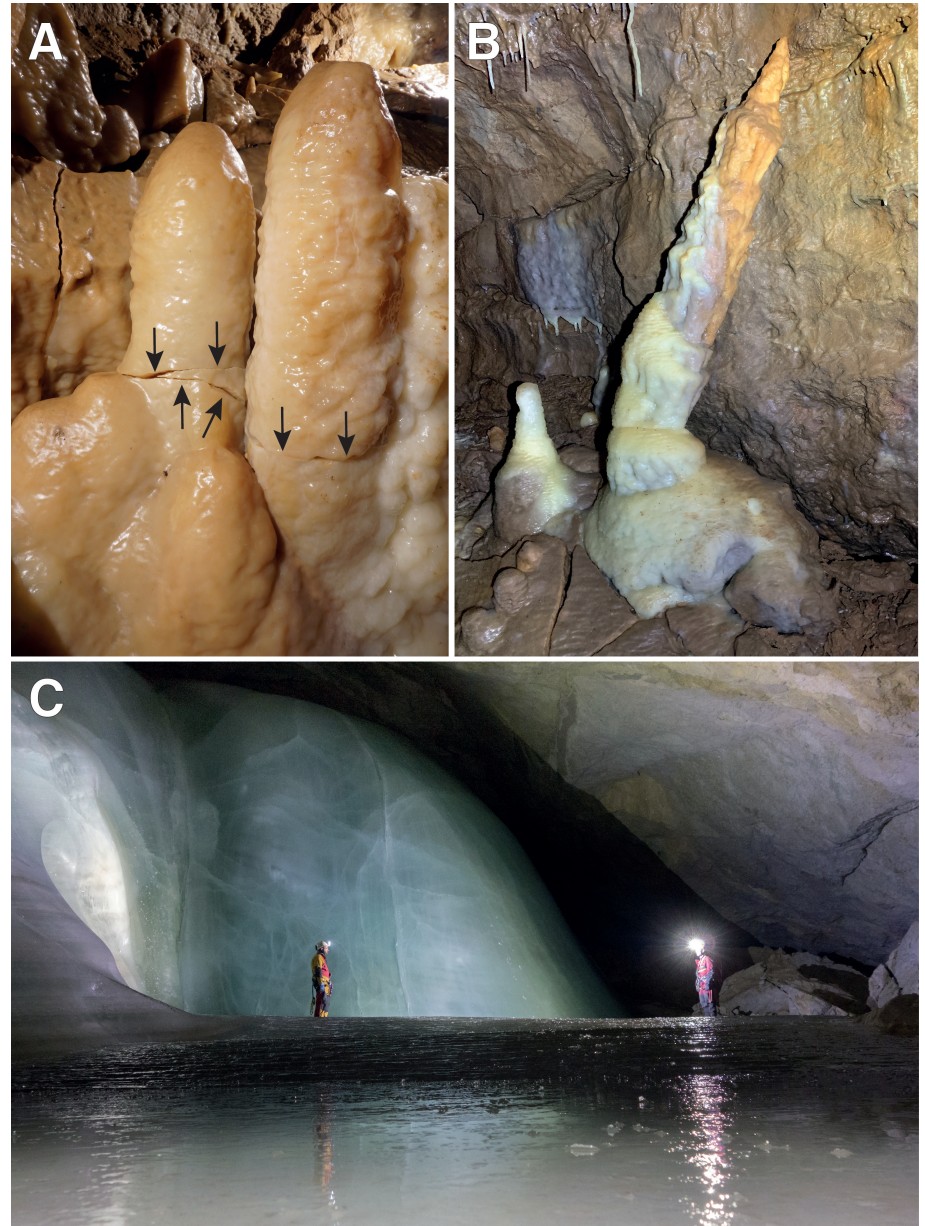

**Figure 1.** A-B: Examples of stalagmites damaged by ice during the last glacial period as shown by the presence of cryogenically formed calcite deposits in these caves. A: Two parallel stalagmites detached from their base along subhorizontal cracks (arrows), Obir Caves, Austria. Width of image 0.4 m. B: Light brown stalagmite broken near its base and tilted. Younger white calcite (of likely Holocene age) is growing over the lower part of this damaged speleothem and forms a smaller stalagmite to the left. Also note dark brown flowstone in the foreground likely fractured due to the former presence of ice. Shatter Cave, Mendip Hills, UK. Width of image 1.1 m. C: Example of a modern ice cave with floor ice and thick ice deposits on the slope in the background. Feuertal Eishöhle, Austria. Images: C. Spötl (A-B), H. Zeitlhofer (D).

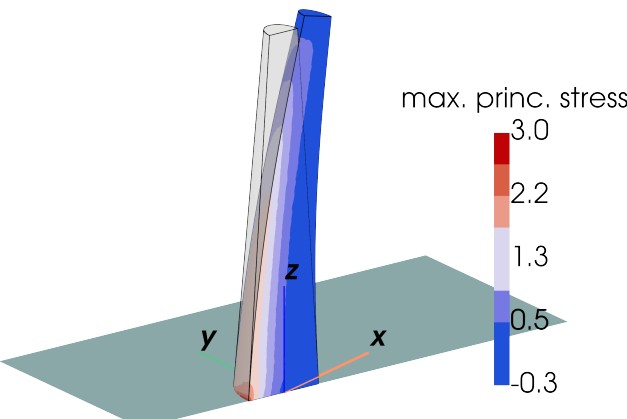

**Figure 2.** Maximum principal stress $\sigma_{\mathrm{max}}$ for the conical stalagmite of simulation 13 in $10^6\,\mathrm{Pa}$ (displacements are scaled by a factor of 1000). Red colours on the upstream face indicate tensile stresses ($\sigma_{\mathrm{max}} > 0$). The $z$-axis of the coordinate system is aligned vertically upward against gravity, while the $x$-axis aligns with the downstream direction of the ice flow.

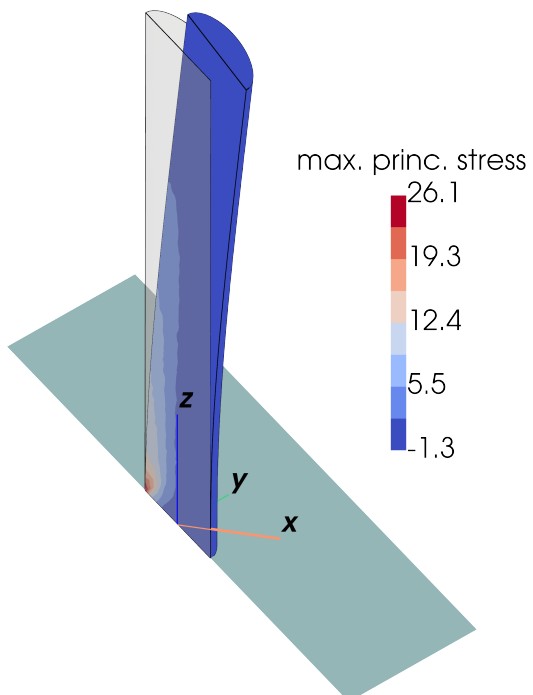

**Figure 3.** Maximum principal stress $\sigma_{\mathrm{max}}$ in the stalagmite pertaining to simulation 09 in $10^6\,\mathrm{Pa}$ (displacements are scaled by a factor of 100). This simulation indicates stresses significantly exceeding the tensile strength of the stalagmite on its upstream side. The $x$-axis points in the downstream direction of the ice flow.