# Peer review of "Brief communication: Stalagmite damage by cave-ice flow quantitatively assessed by fluid-structure-interaction simulations"

_EGUsphere, 2024_

## Author Comment (AC1)

**Author's response to review comments posted for manuscript egusphere-2024-751**

**Date 22.07.2024**

Dear reviewers and dear editor,

we highly appreciate you both having taken time to review our manuscript and sent us valuable comments. We have tried to incorporate all your suggestions into the manuscript. Below we have compiled our point-by-point replies to all your comments in one document.

**Comments of anonymous referee #1 (21.05.2024):**

Reviewer comment:
*This short report is an extension of the modelling described in Spotl et al. 2023, wherein fracture of cave stalagmites by cave ice processes was shown to be more likely caused by thermal expansion of the ice rather than by direct movement of the ice. The research reported here adds more conditions to the basic modelling, and shows that thermal expansion is confirmed as the most likely cause of stalagmite fracture and only in rare situations could fracture be attributed to ice flow dynamics.*

*The techniques used in the modelling appear to be sound and the conclusions justified. The paper is well written and clear. However, there is one glaring ambiguity in the examples shown to illustrate the process and asserted to be "examples of stalagmites damaged by ice during the last glacial period".*

*If a clear example is to be presented, then it must have ice action as the obvious and, ideally, only process that might have caused shattering. Figure 1 B and C are from Shatter Cave, Mendip, UK. Yes, this cave shows nice examples of cryogenic cave calcite deposits, so, yes, ice was present. However, attribution of the cause of the stalagmite shattering is very much compromised by the fact that the cave was only discovered, in 1969, as a result of quarrying, which started in the 1920s. It was named to commemorate the damage assumed to have been done by blasting. If this cave is to be used to support the idea of cryogenic fracturing, then I suggest it be made clear how the cryogenic fracturing differs from quarrying fracturing.*

*Secondly, the assertion of the timing (that the shattering had occurred during the last glacial period) requires more proof. These examples show no obvious post-shattering cementation with calcite and no dates are offered.*

Authors response:
The reviewer is right in pointing out that Shatter Cave also shows speleothem damage due to the blasting activity in this now abandoned quarry. This anthropogenic damage, however, is confined to the

near-entrance part of this cave, where fresh looking fractures are indeed common. Once entering the inner part of this cave, these freshly fractured features disappear and many stalagmites (as well as some flowstones) show evidence of much older damage. The fact that these stalagmites are spatially associated with the presence of cryogenic carbonates (CCC for short) and that many of these formations show "healing" by younger layers of calcite indicates that they are very likely related to the former presence of ice. Unpublished U-Th dates obtained by our colleague Gina Moseley provide evidence that these CCC are late Pleistocene in age.

In response to the reviewer's comment we changed the two images of Shatter Cave and replaced them by another one from this cave where white young calcite coats the damage, clearly showing the speleothem damage is much older than the quarrying. We do not have U-Th dates on this post-damage calcite from this cave, but this type of (locally actively forming) white calcite is well known and characterized in many other caves as Holocene in age.

We would like to express our gratitude for all the valuable comments.

Kind regards,
Alexander Jarosch on behalf of the authors

---

## Author Comment (AC2)

**Author's response to review comments posted for manuscript egusphere-2024-751**

**Date 22.07.2024**

Dear reviewers and dear editor,

we highly appreciate you both having taken time to review our manuscript and sent us valuable comments. We have tried to incorporate all your suggestions into the manuscript. Below we have compiled our point-by-point replies to all your comments in one document.

**Comments of anonymous referee #2 (01.07.2024):**

Reviewer comment:
*My main suggestion for this paper to be published in The Cryosphere would be more context. Although this is a brief communication, as this is not a journal where many readers are going to have knowledge of speleotherms, why this work matters needs to be made much more explicit. This is especially the case given the previous work it builds on had already ruled out ice flow as a mechanism, why did this justify looking at this in more detail, and still getting a result that the process here is likely to lead to fracture? Why is mechanical damage to stalagmites important?*

Authors response:
Thank you for suggesting to put this study into a wider context. We have rewritten and expanded the introduction, providing more background and giving more detail as to why our research matters. Also we have added text that argues for this specific study, namely the combined study (fluid-structure-interaction) of stress created by the flow of ice and potential stress damage caused within the stalagmites in one computer simulation. The previous study only modelled wall shear stress at the stalagmite-ice interface and did not account for stress concentration within the stalagmite, which, as shown in this paper, can cause failure in extreme cases.

Reviewer comment:
*Line 20: How do these relate to stalagmite sizes in previously glaciated regions? Are these typical sizes, and are the new smaller sizes included here also typical?*

Authors response:
The stalagmite dimensions are typical for stalagmites found in caves. So we chose those dimensions to create simulations comparable to real stalagmites.

Reviewer comment:
*Line 23: Do you use a sliding law here? If so please specify and cite.*

Authors response:
No we do not use any sliding laws as we want to study the two end member cases described in the manuscript (frictionless full sliding vs. no-slip). We have added a sentence to make this clear to the reader.

Reviewer comment:
*Line 30: Did you do any sensitivity testing on any of the parameters? E.g. recent studies have shown that the assumption n=3 in Glen's flow law does not always hold (for example see Millstein et al. (2022) https://www.nature.com/articles/s43247-022-00385-x)*

Authors response:
No we did not study the sensitivity of Glen's n on our results, however we are are aware of the literature discussion. Thank you for raising that point. The study you mention is carried out on Antarctic ice shelves. Also Cuffey & Paterson (2010) discuss at length the uncertainties of the Glen ice flow parameters. We simulate ice with 1 and 2 m thickness, so we are far away from the shear magnitudes of Antarctica and even thick alpine-style glaciers. Thus we think our assumption of n=3 is okay. Nevertheless we tested for A being either for temperate ice or at -5 deg C. The differences between the results were less than one-tenth of a percent, thus we ignored these variations and assume A for temperate ice in the study. We have stated this in #54.

Reviewer comment:
*Line 32: Citation needed for this choice of linear elastic.*

Authors response:
We added text and two citations for this choice.

Reviewer comment:
*Line 47: Rather than listing simulation numbers which mean little to the reader this paragraph could highlight better the conditions of the simulation that do find failure.*

Authors response:
Thank you for this comment. You are right, only listing simulation numbers would not make sense. However in the next sentence in the manuscript we actually discuss the conditions for all those run's. Thus we think the manuscript is good as it is.

Reviewer comment:
*Line 66: Is this realistic given in some simulations you're assuming a full-slip condition at the bed, will ice be fully frozen to the stalagmite? You later go on to say full-slip conditions are highly unlikely but if they are worth considering here then why not also for the stalagmite?*

Authors response:
You are right, in reality there might be partial slip on the stalagmite wall, especially if there is basal sliding within the cave. However looking at the stalagmites in Fig. 1, you will see that their walls exhibit quite some geometrical features and thus sliding along the stalagmite walls is rather unlikely. In addition, the direct stress coupling between the ice body and the stalagmite wall is also a model assumption.

Reviewer comment:
*Line 70: Please add more detail on why these values are an overestimation.*

Authors response:
We have added text in the manuscript that describes the situation in detail. As such high stresses would form within the ice downstream of the stalagmite, the ice itself would fail and form cracks. Such cracks would reduce the ability to transfer "pull" of the ice to the stalagmite. As our simulations do not account for crack formation, we transfer the full stress to the stalagmite, resulting in an overestimation of the stress formed within the stalagmite.

Reviewer comment:
*Line 94: Could a combination of these processes be occurring?*

Authors response:
A combination of both processes is theoretically possible. However the highly-varying cave geometries (roughness and steepness of cave floor, cave geometry around stalagmite) found in reality require a detailed case-by-case study to identify the actual failure mechanisms. In this contribution we merely describe the principle mechanisms possible and what circumstances would favor which process.

We would like to express our gratitude for all the valuable comments.

Kind regards,
Alexander Jarosch on behalf of the authors

---

## Author Response (AR1)

**Author's response to editors comment posted for manuscript egusphere-2024-751**

**Date 08.06.07.2024**

Dear Nana B. Karlsson,

Thanks you for taking the time to again review our manuscript and suggest valuable changes.

Editors comment:

*In the response to referee #1, you write "We do not have U-Th dates on this post-damage calcite from this cave, but this type of (locally actively forming) white calcite is well known and characterized in many other caves as Holocene in age."*
*I am aware that you cannot share unpublished data. Instead, I ask that you include a brief sentence about the studies you mention where stalagmites have been dated including appropriate references.*

*I agree with referee #2 that listing the numbers of the simulations is not helpful. In fact, I would suggest that you remove lines 45-46 since they don't seem to be needed.*

Authors response:

We have added a sentence in line #20 (new manuscript) to expand on the concept of post damage calcite formation and how it relates to the age of the damage. Also we have added two references to support this.

We have removed lines #47-48 in the initially submitted manuscript as requested by you. We think you refereed to these two lines instead of lines #45-46.

The point by point reply to the reviewers comments has already been submitted and should be available within the Copernicus system.

Kind regards,
Alexander Jarosch on behalf of the authors